# A Metabolomics Workflow for Analyzing Complex Biological Samples Using a Combined Method of Untargeted and Target-List Based Approaches

**DOI:** 10.3390/metabo10090342

**Published:** 2020-08-25

**Authors:** Thomas Züllig, Martina Zandl-Lang, Martin Trötzmüller, Jürgen Hartler, Barbara Plecko, Harald C. Köfeler

**Affiliations:** 1Core Facility Mass Spectrometry, Medical University of Graz, 8036 Graz, Austria; thomas.zuellig@medunigraz.at (T.Z.); martin.troetzmueller@medunigraz.at (M.T.); 2Department of Paediatrics and Adolescent Medicine, Division of General Paediatrics, University Childrens’ Hospital Graz, Medical University of Graz, 8036 Graz, Austria; martina.zandl@medunigraz.at (M.Z.-L.); barbara.plecko@medunigraz.at (B.P.); 3Institute of Pharmaceutical Sciences, University of Graz, 8036 Graz, Austria; juergen.hartler@uni-graz.at

**Keywords:** LC-MS, metabolomics, LDA, XCMS, plasma, CSF

## Abstract

In the highly dynamic field of metabolomics, we have developed a method for the analysis of hydrophilic metabolites in various biological samples. Therefore, we used hydrophilic interaction chromatography (HILIC) for separation, combined with a high-resolution mass spectrometer (MS) with the aim of separating and analyzing a wide range of compounds. We used 41 reference standards with different chemical properties to develop an optimal chromatographic separation. MS analysis was performed with a set of pooled biological samples human cerebrospinal fluid (CSF), and human plasma. The raw data was processed in a first step with Compound Discoverer 3.1 (CD), a software tool for untargeted metabolomics with the aim to create a list of unknown compounds. In a second step, we combined the results obtained with our internally analyzed reference standard list to process the data along with the Lipid Data Analyzer 2.6 (LDA), a software tool for a targeted approach. In order to demonstrate the advantages of this combined target-list based and untargeted approach, we not only compared the relative standard deviation (%RSD) of the technical replicas of pooled plasma samples (*n* = 5) and pooled CSF samples (*n* = 3) with the results from CD, but also with XCMS Online, a well-known software tool for untargeted metabolomics studies. As a result of this study we could demonstrate with our HILIC-MS method that all standards could be either separated by chromatography, including isobaric leucine and isoleucine or with MS by different mass. We also showed that this combined approach benefits from improved precision compared to well-known metabolomics software tools such as CD and XCMS online. Within the pooled plasma samples processed by LDA 68% of the detected compounds had a %RSD of less than 25%, compared to CD and XCMS online (57% and 55%). The improvements of precision in the pooled CSF samples were even more pronounced, 83% had a %RSD of less than 25% compared to CD and XCMS online (28% and 8% compounds detected). Particularly for low concentration samples, this method showed a more precise peak area integration with its 3D algorithm and with the benefits of the LDAs graphical user interface for fast and easy manual curation of peak integration. The here-described method has the advantage that manual curation for larger batch measurements remains minimal due to the target list containing the information obtained by an untargeted approach.

## 1. Introduction

Metabolomics is defined as an “omics” approach in which the global set of small molecules (<1.5 kDa) in a distinct biological sample, e.g., tissue, blood, cerebrospinal fluid (CSF), cells or their compartments, are analyzed with the prospect of a better understanding of their biological function [1]. Metabolomic approaches to clinically relevant data include differential analysis between control groups and diseased groups, which can lead to the identification of novel biomarkers reflecting those abnormalities. Various potential biomarker candidates have been introduced in several clinical fields including respiratory [2,3], neurological [4], and kidney diseases [5,6], diabetes [7,8], cancer [9,10], and metabolic disorders [11]. Depending on the underlying hypothesis, different approaches are applied in metabolomics research. On the one hand there are established targeted workflows where samples are screened for a set of specific and pre-defined, metabolites (e.g., amino acids, biogenic amines, nucleobases, organic acids, sterols, organic phosphates, and vitamins) [12]. Untargeted workflows on the other hand, are non-hypothesis-driven, allowing deeper insights into complex biological samples and, therefore, may also lead to the discovery of new metabolic hypotheses as well as novel biomarkers. The challenge of untargeted metabolomics in larger sample cohorts is the time-consuming data processing, the annotation of unknown compounds and their verification [13,14,15]. In the field of mass spectrometry (MS) metabolomics, MS devices are usually coupled with an additional separation step—except for direct infusion techniques—to reduce sample complexity and to increase coverage and detection sensitivity [16]. The most used coupling technique is liquid chromatography (LC). This technique benefits from a variety of columns with different stationary phases, which allows separating a large number of distinct polar compounds. Frequently used columns are reversed phased (RP) C18 or C8 [17,18] and hydrophilic interaction chromatography (HILIC) [19]. The variety of these columns offers additional information depending on the composition of the stationary phase and the selected pH range [20]. Gas chromatography (GC) is especially used in breath metabolomics, as shown in several studies [21], benefitting from superior separation and less signal-to-noise ratio compared to liquid chromatography (LC). In addition, GC-MS is used beyond volatile organic compounds by applying different derivatization techniques [22,23]. Capillary electrophoresis coupled to MS (CE-MS) is a beneficial method for highly polar compounds of the metabolome, albeit not as popular as the chromatography techniques [24]. For processing of obtained MS data several software tools are available, e.g., XCMS [25], MZmine [26] or OpenMS [27]. As diverse as the software pool is, results can vary widely and data processing tools have to be adapted to individual biomedical applications [28,29]. Several methods for improving annotation and data quality are available using algorithms with false discovery rate [30], frameworks based on machine learning methods which can improve the quality of automated data processing workflows [31], or a sophisticated graphical interface for customizing parameters to reprocess data [26].

The aim of this work was to demonstrate selective separation of metabolites by performing a HILIC approach within a comprehensive metabolomics workflow. Therefore we used two types of software, Compound Discoverer 3.1 for untargeted metabolomics and Lipid Data Analyzer 2.6 for a target-list based approach to benefit from LDA 3D algorithm for more precise peak border confinement. We also compare the performance of this combined approach with typical untargeted metabolomics software tools (CD and XCMS online), for which we have used the precision of repeatedly measured pooled serum and pooled CSF samples. Since annotation of compounds for metabolomics studies can be complex, we have applied a level of trust that was introduced by the Metabolomics Standards Initiative (MSI).

## 2. Materials and Methods 

### 2.1. Pooled Serum and Pooled CSF Samples

Pooled CSF and pooled plasma samples were obtained from six female patients suffering of Rett Syndrome and 13 (eligible for pooled CSF) and 25 (eligible for pooled plasma) healthy children, respectively. All individuals were aged between 4 and 16 years. The pooled samples were used to develop and evaluate this metabolomics workflow. The study itself is an ongoing project and not the objective of this publication. The project is ethically approved (EK-number 31-162ex18/19).

### 2.2. Chemicals

Acetone (34850-2.5L-M; ≥99.8%) from Sigma-Aldrich (St. Louis, MO, USA), and acetonitril chromasolv (34851-2.5L; ≥99.9%), and methanol chromasolv (34885-2.5L; ≥99.9%) from Honeywell (Charlotte, NC, USA).

### 2.3. Metabolite Standards

Dopamine HCl (56610-5G; ≥98.5%), l-ornithine monohydrochloride (75469-25G; ≥99.5%), l-prolin (81710-10G; ≥99%), taurine (86330-25G; ≥99%), creatinine (C4255-10G; ≥98%), d-(+)-glucose (G8270-100G; ≥99.5%), l-alanine (A7627-1g; ≥98%), l-asparagine (A0884-25G; ≥98%), l-aspartic acid (A8949-25G; ≥99%), l-citrulline (C7629-10MG; ≥98%), l-glutamic acid (G1251-100G; ≥99%), l-glutamine (G8540-25G; ≥99%), l-histidine (H8000-5G; ≥ 99%), l-isoleucine (I2752-1G; ≥98%), l-leucine (L8000-25g; ≥98%), l-lysine (L5501-1G; ≥98%), l-methionine sulfoxide (M1126-1G; NA), l-phenylalanine (P2126-100G; ≥98%), l-serine (S4500-1G; ≥99%), l-threonine (T8625-1G; ≥ 98%), l-tryptophan (T0254-5G; ≥98%), l-tyrosine (T3754-50G; ≥98%), l-valine (V0500-1G; ≥98%), cholic acid (C1129-25G; ≥98% ), d-carnitine (544361-1G; ≥98%), decanoyl-l-carnitine (50637-10MG; ≥94%), Folic acid (F7876-1G; ≥97%), hippuric acid (112003-5G; ≥98%), l-carnosine (C9625-10MG; ≥99%), Palmitoyl-l-carnitine (91503-10MG; ≥95%), Riboflavin (R4500-5G; ≥98%), valeryl-l-carnitine (04265-10MG; ≥95%), adenine (A8626-1G; ≥99%), cytidine (C122106-1G; ≥99%), cytosine (C3506-1G; ≥99%), d-arginine (A2646-250MG; ≥98%), choline (C7017-10MG, ≥99%), a-tocopherol (T3251-5G, ≥96%), adenosine (A9251-1G, ≥99%), and methionine (M9625-5G, ≥98%) standards were purchased from Sigma-Aldrich (St. Louis, MO, USA), and Estradiol (E0950-000; NA) from Steraloids (Newport, RI, USA).

### 2.4. Sample Preparation and Storage

Sample preparation was performed according to Bruce et al. [32]. Briefly, proteins were precipitated by adding a 3:1 volume of ice-cold acetonitrile/methanol/acetone (1/1/1, *v*/*v*/*v*) to 50 µL pooled plasma or 100 µL pooled CSF samples and by vortexing these for 15 s. After a precipitation step at 4 °C for 60 min, samples were centrifuged at 12,419 relative centrifugal force (rcf) for 10 min (Hereaus Biofuge pico, Hanau, Germany). The resultant supernatants were aspirated into clean Eppendorf tubes. The samples were concentrated under a stream of dry nitrogen gas at room temperature (Techne, FSC400D, Staffordshire, UK). The resultant plasma and CSF sample pellets were re-suspended in acetonitrile/water (1/1, *v*/*v*) to 50 µL sample volume, respectively, and immediately stored at −80 °C until further analysis. In order to evaluate data processing and annotation a mixture of metabolite standards (shown in Section 2.3) were measured in addition to biological samples. The final injection concentration was 10 µM solved in acetonitrile/water (1/1, *v*/*v*). 

### 2.5. LC-MS/MS Method

For all samples a full-scan mass-spectrometric interrogation of each sample’s small molecule was achieved by Dionex Ultimate XRS UHPLC (ultra-high performance liquid chromatography)-Orbitrap Velos Pro hybrid mass spectrometer (Thermo Fisher Scientific, Waltham, MA, USA). Separation was performed on an Acquity UPLC BEH Amide column (2.1 mm × 150 mm, 1.7 µm) (Waters Corporation, Milford, USA), thermostated to 40 °C. Mobile phase A was 97% ACN + 3% H_2_O + 0.1 mM NH_4_COOH + 0.16% HCOOH, Mobile phase B was H_2_O + 0.1 mM NH_4_COOH + 0.16% HCOOH, and autosampler wash solution ACN/H_2_O (1/1, *v*/*v*). The starting point of gradient elution was 5% mobile phase B and increased up to 30% over 30 min. Mobile phase B was reset to start conditions over a minute and re-equilibrated for 9 min before 2 µL of the next sample was re-injected. Flow rate was 200 µL min^−1^ and samples were thermostated at 8 °C in the autosampler.

The Orbitrap Velos Pro operated in data dependent acquisition mode using a HESI II ion source. Full scan spectra from *m*/*z* 60 to 1600 were acquired in the Orbitrap with a resolution of 100,000 (*m*/*z* 400) in positive mode and the 10 most abundant ions of the full-scan spectrum were sequentially fragmented with CID (normalized collision energy, 50) and analyzed in the linear ion trap. Isolation width was 1.5, activation Q: 0.2; activation time: 10, and the centroided product spectra at normal scan rate were collected. The exclusion time was set to 11 s and as lock mass a polysiloxane with *m*/*z* 536.16537 was chosen. 

The following source parameters were used: Source voltage: 4.5 kV, source temperature: 275 °C, sheath gas: 25 arbitrary units, aux gas: eight arbitrary units, sweep gas: zero arbitrary units, capillary temperature: 300 °C. 

### 2.6. Data Processing

The data processing workflow was split into two parts (Figure 1). The compound discoverer 3.1 (CD) [28] was used for untargeted and novel feature detection and annotation with library scoring (Figure 1, left side). Previously generated target lists were merged and used to process data with the Lipid Data Analyzer 2.6 (LDA) software tool (Figure 1, right side). 

Raw files were processed with CD in the following steps: First, spectra were selected from raw data followed by retention time (RT) alignment with RT tolerance of 1 min and 5 ppm mass precision. Then unknown compounds were detected and grouped with RT tolerance of 0.7 min and 5 ppm mass deviation. In a second step missing values were filled and the compounds were annotated with different types of databases. mzCloud was used to annotate compounds on MS/MS level with a mass tolerance of 10 ppm. Chemspider, which integrates BioCyc [33], Human Metabolome Database [34], and KEGG database [35], was used to annotate features based on exact mass with a mass tolerance of 5 ppm as well as the CD internal database (an endogenous metabolites database of 4400 compounds). The data set obtained was used to generate a list of potential targets taking into account the level of confidence in annotations implemented by the Metabolomics Standards Initiative (MSI) of the Metabolomics Society (Table 1) [14,36]. In detail, the confidence level of annotation was created as follows: In the first step, all hits from the mzCloud—MS/MS scoring database—greater than 80% score were exported. Due to known RT and better fragmentation spectra, the samples found in the list of purchased standards belong to confidence level 1, compounds detected in addition to the purchased standards to confidence level 2. Compounds detected on the basis of the monoisotopic mass in Chemspider or the internal databases were defined as confidence level 3. Unknown compounds of the analyted CSF or plasma samples were classified as confidence level 4. The list of detected compounds included exact molecular mass, RT, compound names, if known, and were completed with purchased standards not found by the untargeted approach. 

The list was manually curated before it was used as a target list for processing with LDA software tool [37,38]. For data processing we disabled isotopic quantitation as CD already used isotopic pattern matching. After processing, extracted ion chromatograms (EICs) for all compounds were exported and corrected manually if necessary. False positive, noise, or chromatographically not resolved compounds were removed from the final target list. To compare results data were independently processed with the XCMS Online platform. Therefore, raw data sets were uploaded to XCMS Online and a single group job was created to analyze the same pooled plasma and pooled CSF samples as already described with CD and LDA. We selected the predefined workflow settings for Orbitrap (Appendix A). The CentWave algorithm was used for features detection, the OBI-warp algorithm for RT correction, CAMERA for annotation of isotopes and adducts, and the METLIN database was used for MS/MS score matching to identify compounds.

## 3. Results

### 3.1. Chromatographic Separation of Metabolites

For development of chromatographic separation we used a set of 41 purchased reference standards, representing a selection of different compound classes found in the human metabolome [39] i.e., amino acids (*n* = 21), biogenic amines (*n* = 5), sugars (*n* = 1), acyl carnitines (*n* = 4), nucleobases (*n* = 4), organic acids (*n* = 1), sterols (*n* = 2), and vitamins (*n* = 3). Since ionization and chromatographic separation strongly depend on functional groups, we observed that these compounds are a good representative set for testing the LC separation capability of the used HILIC BEH amide column in combination with eluents and gradient elution. Due to the different chemical properties of different metabolites, separation of all compounds is challenging. Total ion chromatogram (TIC) (Figure 2, top) and extracted ion chromatograms (EIC) of each reference standard (Figure 2, bottom) demonstrated good separation of the purchased reference material by applying the setup as explained above (Section 2). All standards were detectable and separated either by chromatography, including isobaric compounds such as leucine and isoleucine, or by their mass-to-charge (*m*/*z*) ratio in the mass spectrometric dimension.

### 3.2. Precision Evaluation of LDA Compared to CD and XCMS Online

A key element in finding differences in biological samples is efficient and reliable data processing. By measuring pooled samples, which are technical replicas, we would expect the relative standard deviation to be ideally zero. Due to sample stability, instrument drift, or ionization variability, and other batch effects, this is generally not the case. To demonstrate the analytical concept of an untargeted metabolomics approach using CD and transforming it into a target list-based metabolomics approach using LDA we analyzed pooled plasma (*n* = 5) and CSF (*n* = 3) samples. These pooled samples were repeatedly measured—QC *n*, extracted blank *n*, 10 biological samples, QC *n* + 1, extracted blank *n* + 1, followed by 10 biological samples, and so on—between a longer sequence to simulate the acquisition of a typical larger metabolomics study with a QC-based normalization strategy for regression of systematic errors [40,41]. This takes into account sample stability, instrument drifts, and other errors. For evaluation of the difference between the combined approach of the LDA and CD, and XCMS online, we used the relative standard deviation in percent of the detected compounds. Features that were only detectable in XCMS online or with CD and could not be confirmed with the LDA were not evaluated (Figure 3, Appendix A). Within the pooled plasma samples XCMS online identified 2510 and CD identified 1880 features. Within the pooled CSF samples XCMS online identified 1446 features and CD detected 1005 features. After adduct filtering and isotope pattern recognition, CD detected 1119 compounds in pooled plasma samples and 495 in pooled CSF samples. These results were used in combination with the list of measured purchased standards to process the raw data with LDA. After reprocessing, we identified 580 compounds in pooled plasma samples and 94 compounds in pooled CSF samples. The lower number of compounds in the CSF samples is partly due to the limitation of the sample volume. Concentration of the CSF sample in less injection solvent may result in higher signal intensity. Variation of the %RSD between the combined approach using both CD and LDA compared to CD and XCMS strongly depends on the raw data quality. The difference between pooled CSF samples with signals of lower intensity shows a stronger effect than pooled plasma samples with overall better signal intensity (Figure 3). The results of the pooled plasma samples with a %RSD of less than 25% showed that 68% of the compounds processed with LDA, 57% of the compounds processed with XCMS Online and 55% of the compounds processed with CD fall into this category. The differences between CSF samples with less than 25% %RSD were even greater. In the processed data with LDA it was 83%, with XCMS Online, 28%, and with CD, 8% of the compounds with a %RSD of less than 25%.

### 3.3. Metabolite Validation

In many MS based metabolomics studies, the annotation of features can be misleading because not all information is equally valid. For this reason, the 2005 founded Metabolomics Standards Initiative (MSI) has the aim of establishing reporting standards which are agreed upon by the community. One point was the definition of the quality level for the identification of metabolites [42]. Blaženovic et al. published an extended version with one additional category with confidence levels from 0 to 4 [14]. These definitions clarify how trustworthy the annotation is. The following example shows how these criteria affect the annotation of the compound pipecolic acid (*m*/*z*, 130.0863, M+H) at different confidence levels. Two-hundred nineteen compounds were revealed with a METLIN database search of pipecolic acids precursor mass with a mass tolerance of 5 ppm, which corresponds to a confidence level of 3. With the additional MS/MS spectral information, four compounds were found, two were l- and d-pipecolic acid and the other two were cycloleucine and nipecotinic acid, which represents confidence level 2. Additional reference material would reduce the potential compounds to l- and d-pipecolic acid as the isomer structure could not be elucidated with this method and would be in the second-best category of confidence level 1. This example is limited by the METLIN-database search and the numbers beyond METLIN annotated metabolites can be even greater.

In the pooled plasma 30 compounds were detected with confidence level 1 and, in the pooled CSF samples, there were 23 compounds with level 1 confidence. For these, standards were purchased and RT and fragmentation patterns were known. At confidence level 2, 26 compounds were detected in the pooled plasma and nine in the pooled CSF samples. For them, we used the fragmentation patterns annotated in mzcloud. The rest of the detectable compounds was within confidence level 3 or 4 (plasma 522, CSF 62 compounds). Monoisotopic masses found in the databases explained differences between levels 3 and 4. Despite the indistinct annotation quality of compounds with confidence levels 3 or 4, differential studies are further needed for analysis and comparison, e.g., if these compounds are up- or down-regulated.

## 4. Discussion

In this study, we developed a comprehensive method to separate and detect metabolites with high-resolution mass spectrometry in a data-dependent full-scan acquisition method. We compared and evaluated three different data processing workflows, two MS-based metabolomics software tools (CD, XCMS online) and one combination of a MS-based metabolomics software with a target list-based software tool (CD/LDA). A variety of software tools exist in the field of untargeted metabolomics. Misra et al. have compiled a comprehensive list [43]. One of the most common software tools is MS-Dial [44], which, in addition to the DDA mode in combination with liquid chromatography, also enables data-independent acquisition (DIA) or sequential window acquisition of all theoretical fragment ions (SWATH) and provides an MS/MS scoring database to match experimental spectra against spectral libraries, such as MassBank [45] or LipidBlast [42]. MZmine 2 [26] is another frequently used open source software tool for untargeted metabolomics. It offers a graphical user interface (GUI) and has built-in multiple previews to control and modify setting for peak integration. The compound identification is based on a user-defined database with *m*/*z* values with retention times and can be connected directly to several online resources, e.g., PubChem [33], HMDB [34], or METLIN [35,36]. In our study, we used Compound Discoverer 3.0 (Thermo), a commercial software tool, XCMS online, and Lipid Data Analyzer 2.6, both open-source tools. Compound Discoverer 3.0 offers a user-friendly interface and has built-in statistical analysis, e.g., PCA, Volcano plot, S-plot, hierarchical cluster analysis, and also univariate statistics. The disadvantage is that commercial software is expensive and limited to processing vendor-specific files. XCMS online offers a platform for uploading and processing your datasets online, is manufacturer-independent, and has advanced statistical tests and path analyses. Compared to the other two non-targeted software tools, LDA is a target-list based tool. Peak integration is based on a unique 3D algorithm which showed in the lipidomics field superior accuracy and sensitivity compared to mzMine2 and LIMSA [37]. Li et al. compared Compound Discoverer 2.0 [38], MS-Dial, MZmine 2 and XCMS with a standard mix of 970 compounds analyzed with a Q exactive HF [40]. They used Tracefinder 4.0 (Thermo) with a targeted approach as a benchmark to compare the results against the non-targeted workflows. They showed that all untargeted software tools had similar overall performance in recognizing features. XCMS had the highest identification rate, but MZmine 2 had slightly better performance in terms of accuracy and quantification. Compared to the benchmark method with the targeted approach, all untargeted analyzes lack accuracy and quantification. They summarized that impaired quantification of features is mainly due to incorrect EIC peak detection. In order to reduce false positives and increase the number of features, a combined approach with XCMS and MZmine2 was successfully tested [46]. This could also be an option to use more than one non-targeted metabolomics software tool for increasing the number of features to be used as a target-list for processing the data with LDA to increase precision. The question is whether it is worth spending the additional time to find a few additional features. In our opinion, there is no best metabolomics software solution. In this study we benefit from the combination of the strengths of complementary types of software. Another general problem of LC-MS instruments are batch and time drift effects, especially when data acquisition takes a long time, as instruments tend to change in response over time [47,48]. Nowadays, various methods for correcting these effects are known, e.g., the use of multiple internal standards associated with higher costs, or use of pooled QCs, which are measured repeatedly during the acquisition sequence [41,49,50]. However, if algorithms are optimized in respect to find as many features as possible and thereby, the integration of the peak area is difficult to adapt, it is not possible to curate larger data sets manually. This leads to an increased variance, as shown in Figure 3 and affects the identification and removal of outliers, which is needed for proper batch corrections. The here-presented combined approach demonstrates many advantages in peak area integration and data processing, thereby leading to less variance. This reduction offers benefits for further data processing as well as for statistical tests, but has slight disadvantages in the detection and identification of novel compounds, which are caused by the dilution of the pooled QCs in comparison to individual phenotype samples.

This combined approach will be further applied on plasma and CSF samples from a larger ongoing study with control groups of healthy subjects and study groups with neurological pediatric disorders.

In conclusion, this two-step software approach combines the strength of a broad range of compounds detectable by untargeted metabolomics with the highly reliable peak area integration inherent to LDA, which, at the end of the day, results in more accurate up- and down-regulation patterns of expected but also unexpected individual metabolites.

## Figures and Tables

**Figure 1 metabolites-10-00342-f001:**
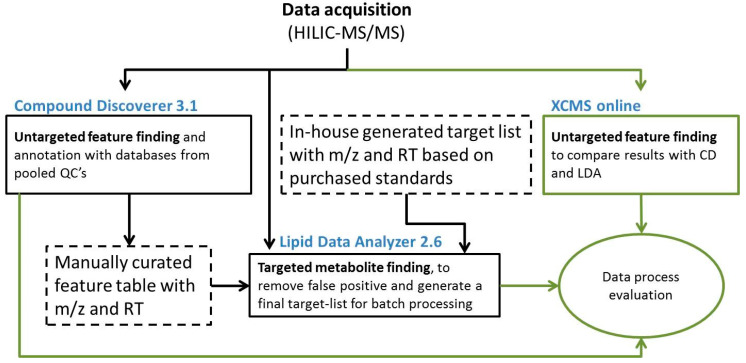
Flowchart of the combined untargeted and target-list based metabolomics workflow used in this study (black) with the extension of data process evaluation of the three software tools (green). Blue titles show software tools used, dashed boxes are target lists based on reference material and an untargeted approach, and boxes with solid lines are the processing steps. (RT: Retention Time; *m*/*z*: mass to charge ratio).

**Figure 2 metabolites-10-00342-f002:**
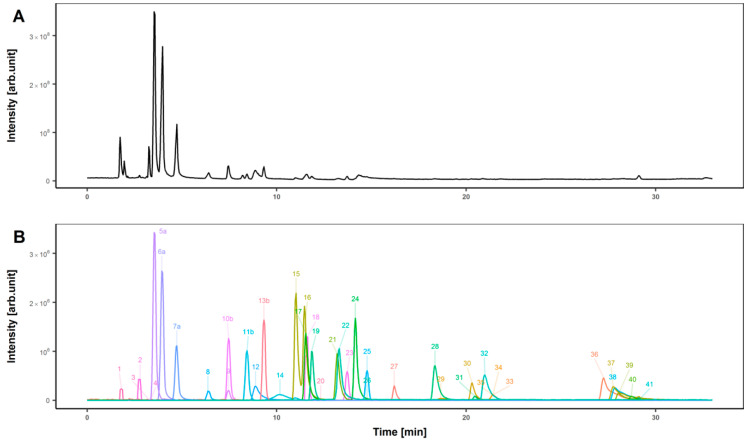
Total ion chromatogram (**A**) of measured reference material with Orbitrap Velos Pro (Thermo). Separation was performed with BEH amide (2.1 mm × 150 mm particle size 1.7 µm) at 40 °C. (**B**) Extracted ion chromatograms (EICs) of identified purchased standards. All standards were injected at a concentration of 10 µM. Due to the different signal abundances, the intensities of six compounds were manually reduced for better illustration (a: with factor 50, b: with factor 5). 1, a-tocopherol; 2, hippuric acid; 3, estradiol; 4, cholic acid; 5a, palmitoylcarnitine; 6a, decanoylcarnitine; 7a, valerylcarnitine; 8, choline; 9, adenine; 10b, adenosine; 11b, creatinine; 12, carnitine; 13b, riboflavin; 14, dopamine; 15, leucine; 16, isoleucine; 17, phenylalanine; 18, cytosine; 19, tryptophan; 20, folic acid; 21, methionine; 22, valine; 23, cytidine; 24, proline; 25, taurine; 26, tyrosine; 27, alanine; 28, threonine; 29, glutamate; 30, glutamine; 31, serine; 32, methionine sulfoxide; 33, asparagine; 34, aspartate; 35, citrulline; 36, arginine; 37, histidine; 38, carnosine; 39, lysine; 40, ornithine; 41, glucose.

**Figure 3 metabolites-10-00342-f003:**
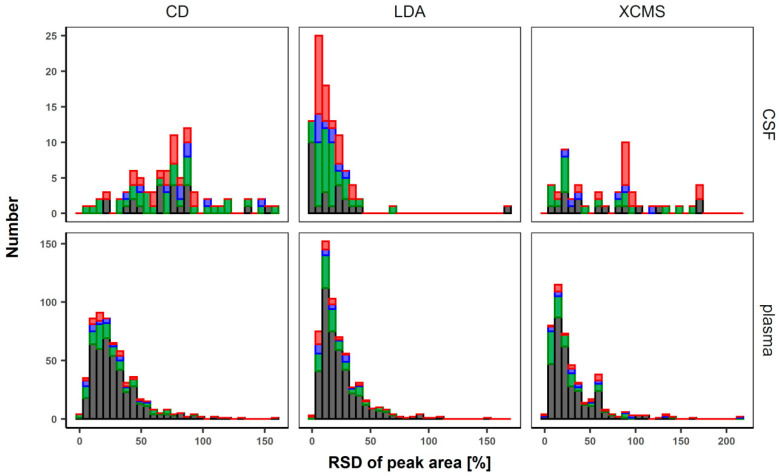
Evaluation of selected and processed data with XCMS online, Compound Discoverer 3.1 (CD) and Lipid Data Analyzer 2.6 (LDA). Relative standard deviations (%RSD) of the peak areas of pooled plasma samples (*n* = 5) and pooled CSF samples (*n* = 3) are shown as a histogram. The color code indicates the confidence level to which the detected compound belongs: Red: group 1, blue: group 2, green: group 3, and black: group 4.

**Table 1 metabolites-10-00342-t001:** Requirements as promoted by the Metabolomics Society [14,36] and their implementation into this study.

Level of Confidence	Description	Data Requirements in This Study	Certainty
4	Unknown feature	Recognized feature with CD and detected with LDA	-
3	Possible structure matched on a single information e.g., database	Matched to the exact mass (5 ppm) in an MS 1 database (Chemspider)	BAD
2	At least two sources of information to match 2D structure. e.g., exact mass and MS/MS score	Matching exact mass (10 ppm) with a fragmentation score over 80 of the mzcould database	GOOD
1	Confidently matched 2D structure with two technics e.g., MS/MS and RT	Exact mass (5 ppm) and experimental MS/MS information and RT information of pure standard	VERY GOOD
0	Compound with determination of the 3D structure	Not possible with this method	BEST

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
