# Peer review of "A Metabolomics Workflow for Analyzing Complex Biological Samples Using a Combined Method of Untargeted and Target-List Based Approaches"

_metabolites, 2020, doi:10.3390/metabo10090342_

Round 1

Reviewer 1 Report

This manuscript describes a combined targeted and untargeted method to detect hydrophilic metabolites from plasma and CSF.

Major points:

  • A major defect in this manuscript is that the objective is not clearly stated. Please state very precisely at the end of introduction the objective of the study. Figure 3 is especially important/ One suggestion is to place it as figure 1 between introduction and results.
  • Figure 3 should be expanded, and XCMS included.
  • Please indicate in the abstract the origin of samples (human? Other species?).
  • Also, describe at the beginning of the materials and methods section the status of the donors (healthy, patient, treatments, etc. ) and general characteristics (at least sex and age). Link this information to the ethical approval reference.
  • RSD acronym is not defined in abstract. Same for CSF.
  • Please summarize in the abstract the results obtained
  • The type of samples, number of samples, should be reminded at the beginning of the 2.1 results section.
  • How the confidence levels are established (figure 2) should be very clearly explained in the main text.
  • The discussion is meagre. The study and the results should be thoroughly put in the context of metabolomics history and currently available methods. Also, there is little or no discussion about the evident differences found between plasma and CSF samples. What are the perspectives for the future? Shall this simple study be validated by larger studies? Is it going to be applied to a particular pathology, group of pathologies?

Minor points:

  • Figure 1 is cited in text as Figure 2
  • Line 83: there is a reference to point 4.2 of material and methods as “above”, but this section is at the end of the manuscript. Please correct.
  • RSD, XCMS, LDA, QC and CD should be defined earlier in the text. A list of abbreviations is strongly recommended to make reading easier. An alternative is that the materials and methods section is placed between the introduction and the results.

Author Response

Please find attached our response to reviewer 1.

Reviewer 2 Report

Please find my comments embedded in the attached manuscript file.

Author Response

Please find attached our response to reviewer 2.

Reviewer 3 Report

The paper from Zullig and Zandl-Lang et al. about a combined approach for data processing of untargeted metabolomics data is very interesting, however a bit laborious, but could serve the purpose if can obtain better RSD.

I have few points that I would like to address:

1) Only QC samples were used for the paper, which means 5 QCs for plasma and 3 for CSF. It is not clear and not specified what the size of the sequence when those QC were included and how many samples/standards were run in between. Furthermore, was the processing done only using those QC samples?

2) The difference in feature detection could also depend on different options or processing optimizations between the XCMS and CD? it would be good to have the parameters used for processing with XCMS.

3) The workflow of the data processing is not very clear...Is it correct that you used only the features from CD for the list of the purchased standards to for the processing with LDA (based on your figure 3)?But then I do not understand the figure 2 with XCMS? Furthermore, when in row 272 you stated that you did some manual curation, was that in respect to what?

4) The metabolite validation/identification step is done on the final set from the LDA processing? And if it is so, it can be added to the Figure 3 and explained in the section.

5) What would it be your consideration in terms of using a not open source software (CD) compared to a free one (XC-MS)?

6) Row 83: I guess you meant below instead of above, plus I think it would be more correct to say section 4 (since includes both the standards and the LC-MS method) 

Author Response

Please find attached our response to reviewer 3.

Round 2

Reviewer 1 Report

The manuscript has been improved significantly.

I only have one minor comment: Table 1 is not cited in the text.

Author Response

Thank you for reading the manuscript so carefully. We now added the reference to Table 1 in the main text on page 5.

Reviewer 2 Report

Reviewer Comments to Authors V2

Please find my comments below as Comment V2:

Response letter to reviewer 2

We thank the reviewer for her/his constructive criticism and encouraging statements. We have changed the manuscript accordingly or, in some cases, explained our position more explicitly below.

Changes made in the manuscript are mentioned below. The numbering of the pages, tables and figures refers to the revised version.

Q: What this CSF means? Has to add here or remove the content in brackets.

A: We have revised the summary and adjusted the abbreviations to make the meaning clear.

Comment V2: Agreed.

Q: Too many abbreviations in abstract like CD, XCMS, RSD. Instead of RSD, I would like you to preferably use %RSD.

A: We changed as suggested RSD to %RSD.

Comment V2: Agreed to the changes made.

Q: Add the full form of LDA here or where the Lipid Data Analyzer word was used first time in abstract. Full name of LDA given at line 133 doesn't make sense because the abbreviation was used more than 10 times before line 133.

A: Full form of LDA is now available in the abstract

Comment V2: Agreed.

Q: How can a protein precipitation method remove interfering protein? The choice of words may be different because even after precipitation the residual solvent may have interference in terms of phospholipids. Since you have used protein precipitation method then there would be possibility of matrix interference at analytes retention time. Could you provide EIC (Extracted ion chromatograms) of blank chromatogram

A: I deleted the interfering as it seems to be misleading. This step is not to remove all interfering compounds, but as proteins are a big part of plasma compound, we remove these to clean up the samples for lesser contaminations on the column and in the MS device.

Comment V2: Now it is more clear to readers. Looks good.

Q: What is the original volume of sample and (Plasma and CSF) and precipitant diluent used?

Q: 0.5uL Is this correct? This is negligible volume. You mean to say 0.5mL?

A: the original volume of sample was 50μl for plasma and 100 μl for CSF and was after precipitation and evaporation re-suspended in a final volume of 50 μl ACN/H20. The information is added, respectively corrected in line 131 and 137

Comment V2: Volume added to the text. I agreed to the corrections made.

I have an additional comment for section 2.4 Sample preparation and storage. Is this extraction method previously reported, or you optimized it for the first time? If this is a previously reported extraction method or modified extraction method, then provide a suitable reference to the text else provide a rational behind choosing this extraction methodology. Do you have extraction efficiency or recovery data? Matrix effect data? How do you know the same extraction methodology is suitable to extract/detect compounds in both the plasma and CSF matrix?

Q: Make (model) of centrifuge, nitrogen evaporator.

A: Model of centrifuge and the concentrator are in the text now available.

Comment V2: Agreed to the changes made.

Q: As per the definition the “Targeted Metabolomics” is a quantitative approach where a group of metabolites are quantitated with the help of (internal or external) compounds. The compounds for which you have the standards for, have been quantified based on internal calibration method while the compounds for which you do not have standards are termed as “putative” and have been quantified using external calibration method using surrogate analyte with same structural & physiochemical properties. In this manuscript since no quantitative analysis was performed, the use of word “Targeted metabolomics” is highly confusing to the readers and must be removed throughout the manuscript and the title has to be reframed. If you wish to keep the word “Targeted metabolomics” then I need quantitative data using multiple reaction monitoring based LC-MS/MS quantification along with validation data. The validation data should include at least partial validation such as Linearity, Accuracy, Precision)

A: For avoiding confusion we changed the term “Targeted Metabolomics” to “target-list based metabolomics” (changes are highlighted). This method is now as you suggested not a targeted approach with quantification, calibration and validation. It is a method for differential analysis of different phenotypic groups with an increased precision because of the use of an target-list based software in combination with an untargeted approach.

Comment V2: Sounds good. Simply changing the term “Targeted Metabolomics” to “target-list based metabolomics” made the meaning very clear. Now there should be no confusion to readers, and validation data is not needed.

Author Response

We used an already existing sample preparation protocol and now added the appropriate reference.